# Comparisons of Effectiveness in Differentiating Benign from Malignant Ovarian Masses between Conventional and Modified Risk of Malignancy Index (RMI)

**DOI:** 10.3390/ijerph20010888

**Published:** 2023-01-03

**Authors:** Charuwan Tantipalakorn, Dangcheewan Tinnangwattana, Thitikarn Lerthiranwong, Suchaya Luewan, Theera Tongsong

**Affiliations:** Department of Obstetrics and Gynecology, Faculty of Medicine, Chiang Mai University, Chiang Mai 50200, Thailand

**Keywords:** benign ovarian mass, malignant ovarian mass, risk of malignancy index, ultrasound

## Abstract

Objective: To compare the predictive performance in differentiating benign from malignant ovarian masses between the modified risk malignancy index (RMI) and the conventional RMI (RMI-1 and RMI-2). Methods: Women scheduled for elective surgery because of adnexal masses were recruited to undergo pelvic sonography within 24 h before surgery to assess the sonographic characteristics of the masses, focusing on loculi, solid part, ascites, bilateralness, papillary projection, and color flow mapping (CFM). Preoperative CA-125 levels were also measured. Modified RMI, RMI-1, and RMI-2 systems were used to predict malignant masses. The gold standard was pathological or intraoperative diagnosis. Results: A total of 342 ovarian masses, benign: 243 (71.1%); malignant: 99 (28.9%), meeting the inclusion criteria were analyzed. The sensitivity and the specificity of the modified RMI (87.9% and 81.9%) were significantly higher than those of RMI-1 (74.7% and 84.4%), and RMI-2 (79.8% and 81.1%, respectively). Based on ROC curves, the area under the curves were 0.930, 0.881 and 0.882 for modified RMI, RMI-1 and RMI-2, respectively. Conclusion: Modified RMI had better predictive performance than the conventional RMI in differentiating between benign and malignant ovarian masses. Modified RMI may be useful to help general gynecologists or practitioners to triage patients with an adnexal mass, especially in settings of low resources.

## 1. Introduction

Preoperative sonographic assessment of adnexal mass to distinguish between benign and malignant tumors is very important for a plan of management and patient counseling, since the approaches of the two entities are different. Malignant ovarian tumors usually need high surgical expertise in treatment or referral to a tertiary center where gynecologic oncologists are available, whereas benign tumors can be successfully operated on by general gynecologists or can be safely selected for laparoscopic surgery. Preoperative ultrasound is a practical, feasible, effective and inexpensive tool widely used to differentiate between benign and malignant ovarian tumors. Several sonographic algorithms have been developed for such a purpose. One of the most common sonographic approaches, which is widely used in low-resource countries, is the risk of malignancy index (RMI) [1,2,3,4]. The RMI is a scoring system based on sonographic features, menopausal status and serum levels of CA-125. Originally, the RMI system (RMI-1) was developed in 1990 by Jacob et al. [5]. They found that RMI-1 had a sensitivity of 85% and that the specificity was 97% in predicting malignancy, when using an RMI cut-off level of 200. In 1996, Tingulstad et al. [6] developed RMI-2 through the modification of RMI-1 to improve predictive performance. They demonstrated that RMI-2 had better performance than RMI-1 in differentiating benign from malignant adnexal masses, giving a sensitivity and specificity of 80% and 92%, compared to 71% and 96% of RMI-1, respectively. In 2008, Timmerman et al. [7] firstly developed International Ovarian Tumor Analysis (IOTA), simple rules to differentiate between benign and malignant ovarian masses, solely based on sonographic features indicating malignancy (M-rules) or benignity (B-rules). However, the IOTA rules seem to be relatively complicated and give an inclusive result in approximately 20% of cases [7,8]. Accordingly, IOTA rules are not widely practiced in developing countries. The conventional RMI system includes five ultrasound parameters (bilateralness, multilocularity, solid area, ascites and intra-abdominal metastasis). In our experience and the reviewed literature, bilateralness or multilocularity has not been shown to have a high power of prediction, whereas papillary projections and high vascularity have high predictive value [9]. Currently, ultrasound available worldwide has the function of color flow mapping (CFM). Accordingly, we developed modified RMI from RMI-2 by replacing bilateralness and multilocularity with papillary projections and high vascularity.

We have developed a prospective database of preoperative ultrasound under our project of adnexal mass imaging for more than ten years [10,11,12]. Therefore, we had a great opportunity to reevaluate the effectiveness of various preoperative predictive systems and develop a new technique with better performance suitable for a primary health care center, based on our recorded ultrasound findings and clinical characteristics. For general concept, the conventional RMI includes some ultrasound features (multilocularity and bilateralness), which we believe are less predictive than the other features (color flow and papillary feature) that are not included in the conventional RMI. Therefore, we modified the RMI by replacing multilocularity and bilateralness with color flow and papillary feature to develop the modified RMI. The objective of this study was to compare the predictive performance of modified RMI and conventional RMI-1/RMI-2, performed by general gynecologists in distinguishing benign from malignant masses.

## 2. Patients and Methods

This study was conducted as a secondary analysis of our prospective database of adnexal mass projects [10,11,12], at Maharaj Nakorn Chiang Mai Hospital, a tertiary center and medical teaching school. The study was ethically approved by the Institute Review Board (Study code: OBG-2565-09162). The study population, included in the database, were patients admitted for elective pelvic operation because of an adnexal mass, between April 2016 and March 2022. On the database creation, the patients were counseled and invited to participate in the project with written informed consent. The inclusion criteria for recruitment of the patients are as follows: (1) the patients were diagnosed with an ovarian tumor or adnexal mass based on a previous pelvic ultrasound study, and (2) the definite diagnosis of the mass was unknown before the operation, either by previous laparoscopy or previous pelvic surgery. The patients included in this analysis underwent transvaginal and/or transabdominal sonography within 24 h of surgery, using a real-time 5–7.5 MHz transvaginal or 3.5–5 MHz transabdominal curvilinear transducer equipped with Voluson E8 or Voluson E10 (GE Medical Systems, Zipf, Austria), using the following protocol: (1) Transabdominal two-dimension (2D)-ultrasound was firstly performed during full bladder, and then, if needed, (2) Transvaginal ultrasound was performed during empty bladder. (3) A grayscale scan was applied to identify the following features: (a) ascites, (b) solid/cystic component, (c) papillary projections, (d) bilateralness, (e) multilocularity, and (f) intra-abdominal metastasis. (4) Color flow Doppler was applied to subjectively score blood flow as no/minimal (grade 1,2) or moderate/strong (grade 3,4), using power or high-definition color Doppler. Sonographic studies were performed by general gynecologists. The ultrasound examiners were blinded to the clinical data of the participants. The details of ultrasound findings were based on 2D real-time and color Doppler ultrasound, including bilateralness, solid part, uni-/multilocularity, ascites, intraabdominal metastasis, papillary structures, size, color flow mapping and acoustic shadow. The baseline and clinical characteristics of the women, such as age, levels of CA-125 if available, etc., and sonographic features were prospectively recorded in the research record forms and stored in the computerized database on the hard drives. The definition of sonographic features followed The Consensus Guidelines from the American College of Radiology (ACR) Ovarian-Adnexal Reporting and Data System (O-RADS) Committee [13].

The RMI scoring system involved the incorporation of sonographic features of the mass, menopausal status and serum levels of CA 125 to score the mass. The ultrasound features (U) used in RMI-1 and RMI-2 were based on the presence of the following features: (1) multi-locular cyst, (2) bilateral masses, (3) solid area, (4) ascites and (5) intra-abdominal metastases. In the case of bilateral masses, only the most complex mass was selected for analysis. Status of menopause (M) was defined as post-menopause for women having their final menstrual period one year or more previously and pre-menopause for those having menstrual periods or a final menstrual period within the last year. The level of serum CA-125 was defined as the level measured within one week prior to surgery. The RMI scoring was computed as follows: ***(1) RMI-1*** was scored according to the system proposed by Jacobs et al. [5]: RMI scores = U × M × CA-125 level (units/mL), giving an ultrasound score (U) of 0 if there were no ultrasound feature as described above, a score of 1 if there was one feature and a score of 3 if there were 2 or more features; premenopausal status (M) giving a score of 1 for pre-menopause and a score of 3 for post-menopause. The CA 125 level was directly multiplied into the equation. ***(2) RMI-2*** was scored according to the system proposed by Tingulstad et al. [6]: RMI scores = U × M × CA-125 level (units/mL), giving an ultrasound score (U) of 0 if there was no no ultrasound feature, a score of 1 if there was one feature and a score of 4 if there were 2 or more features; M giving a score of 1 for pre-menopause and a score of 4 for post-menopause. The CA 125 level was directly multiplied into the equation. ***(3) Modified RMI*** was scored the same as RMI-2, except that ultrasound features item 1 (multi-locular) and 2 (bilateral) were replaced with (1) moderate to strong color flow (color score 3–4) and (2) presence of four or more papillary projections or nodules (solid component whose height >3 mm, arising from the cyst wall or the septum and protruding into the cyst cavity). Examples of sonographic features of the adnexal masses are presented in Figure 1.

The gold standard or final diagnosis of the adnexal mass relied on histological diagnosis or the diagnosis concluded by the operative team based on intra-operative findings in case of no pathological specimens. The final diagnoses were finally categorized into two groups as those with a benign mass and those with a malignant mass. A mass diagnosed with low-malignant potential was considered a malignant mass. The patients who underwent operation beyond one day after sonographic examination and those with no measurement of CA 125 levels before the operation were excluded from analysis.

***Statistical Analysis:*** Predictive performance of modified RMI, RMI-1 and RMI-2 was analyzed to determine sensitivity, specificity and predictive values. A comparison of the diagnostic indices in differentiating benign from malignant ovarian masses among the three methods was performed, using the area under the curve of the receiver operating characteristic (ROC) curve, based on the method of DeLong [14] and Hanley [15]. Statistical procedures were performed using the statistical package for the social sciences (SPSS) software version 26.0 (IBM Corp. Released 2019. IBM SPSS Statistics for Windows, Version 26.0, IBM Corp., Armonk, NY, USA).

## 3. Results

During the study period, a total of 512 women were recruited. Of them, 170 were excluded because of various reasons, as shown in Figure 2. The remaining 342 women meeting the inclusion criteria were included in the analysis. The mean (±standard deviation: SD) age of the women was 41.4 ± 12.4 years (range 12 to 79). Nulliparous women accounted for 47.1% (161 women), and the remaining 52.9% (181) were parous women, with parity ranging from 1 to 12. Most of the women (265; 77.5%) were in the age of pre-menopause and the remainder (77; 22.5%) were in the age of post-menopause. Of 342 adnexal masses, 71.1% (243 cases) were benign tumors and 28.9% (99 cases) were malignant tumors, consisting of 89 cases of cancers and 10 cases of low-malignant-potential tumors, as presented in Table 1. Out of all of them, the endometriotic cyst was the most common mass, appearing in 23.4% of cases, followed by mature teratoma (13.7%). The most common malignancy was endometrioid carcinoma (25 cases: 7.3% of all masses). The final diagnoses of all adnexal masses are categorized and presented in Table 1.

The diagnostic performance of the three methods in predicting malignant ovarian masses is presented in Table 2. The specificity of the three models was comparable (McNemar’s Chi square test; *p*-value 0.136), whereas the sensitivity of modified RMI was significantly greater than RMI-1 and RMI-2 (87.9% vs. 74.7% vs. 79.8%; respectively; McNemar’s Chi square test; *p*-value < 0.001). Based on the ROC curves (Figure 3), the diagnostic performance of the modified RMI was significantly higher than the other two methods (area under the curve: 0.930, 0.881 and 0.882 for modified RMI, RMI-1 and RMI-2, respectively; *p*-value: 0.008 and 0.009 when compared to RMI-1 and RMI-2, respectively). RMI-1 and RMI-2 had comparable diagnostic performance or area under the curve (*p*-value 0.782).

## 4. Discussion

The main objective of this study was to develop a simple, inexpensive, effective and practical tool to help physicians in primary health care centers or low-resource settings effectively triage patients with adnexal masses. The new insight gained from this study is that the modified RMI has higher diagnostic performance in distinguishing malignant from benign masses than does the conventional RMI. The finding suggests that the modified RMI may be useful in clinical practice, especially in primary health care settings. This can help general gynecologists or practitioners triage patients with adnexal masses for further proper management. For example, benign ovarian masses such as a simple serous cyst or mature cystic teratoma may be treated with simple cystectomy by general practitioners in a community hospital, laparoscopic surgery if available or open laparotomy at local hospitals, whereas malignant ovarian tumors usually require advanced surgery for complete surgical staging that need consultations gynecologic oncologists or referral to a tertiary care center. 

The rationale for the development of the modified RMI is based on the fact that (1) Papillary projections and strong color flow in the masses, which are not included in the conventional RMI, are highly predictive [9]. (2) According to several pieces of evidence in recent years, most popular new ultrasound methods with high effectiveness in discriminating the adnexal masses, such as IOTA (International Ovarian Tumor Analysis (IOTA) group) simple rules, Assessment of Different NEoplasias in the adneXa (ADNEX) model and O-RADS, do not include sonographic features of bilateralness and multilocular mass as a main component of the systems [9,13,16,17], probably due to less predictive values. (3) Color Doppler ultrasound is available in most modern simple ultrasound machines as a user-friendly function. Therefore, we developed the modified RMI by replacing bilateral and multilocular masses with papillary projections and strong color flow, in the hope that they might increase the accuracy of prediction. Whereas several tumor markers are helpful in the differential diagnosis of ovarian cancers and used in many predictive systems like Ovarian Malignancy Risk Algorithm (ROMA) Score [18] etc., only CA-125 is available worldwide and suitable for practice in primary health care centers. Therefore, we included only CA-125 in the modified RMI. However, theoretically, serum levels of CA-125 are relatively low in patients with sex-cord tumors or malignant germ cell tumors, but they can simply be missed with conventional RMI. Whereas these tumors are commonly associated with sonographic features of strong color flow or papillary projections arising from the cyst wall, the modified RMI can theoretically improve the sensitivity in cases of solid cancers with low CA-125 levels.

We have developed the modified RMI with the intention of improving the diagnostic performance of the conventional RMI. We expected an improvement of sensitivity and specificity with the modified RMI. It is unclear how the modified RMI can improve predictive performance when compared to conventional RMI. Partly, it can be explained by the fact that the cancers with low CA-125 levels, like malignant germ cell tumors, and with only one ultrasound feature (such as solid mass) resulted in a score lower than 200, leading to a prediction of benign mass, but the detection of strong color flow in the masses can result in a higher score of greater than 200, predicting malignancy and the improvement of sensitivity. Likewise, we noted that one case of clear cell carcinoma was predicted for benign tumor by conventional RMI because of low CA-125 level and only one ultrasound feature of multilocular mass, resulting in a prediction as benign by conventional RMI, but with modified RMI the cases were predicted to be malignant because of small papillary projections arising from the cystic wall with vascularization. On the other hand, many cases of benign masses sometime give false positive conventional RMI, especially bilateral/multilocular mature cystic teratoma with solid areas in the mass with low CA-125 levels, and bilateral/multilocular endometrioma are often associated with high CA-125 levels. With modified RMI, improving specificity or reducing false positives may be expected because of the lack of a sonographic score from bilateralness and multilocularity and also no score obtained from low vascularity in the replacement marker (color flow mapping). Nevertheless, we have noted that the modified RMI did not improve the specificity of prediction. This may be because these tumors had low levels of CA-125 and were likely to have low scores, even when RMI-1 or RMI-2 resulted in an initial prediction of benignity. Thus, the addition of color flow did not improve specificity.

Other than RMI systems, there are several models proposed to discriminate between benign and malignant tumors. The main models for preoperative prediction of ovarian tumors are as follows [16]: (1) IOTA simple rules based on 10 binary sonographic features, including five benign (unilocular cyst, smooth multilocular cyst with largest diameter <100 mm, presence of solid areas with largest diameter <7 mm, acoustic shadows, no vascularization on color Doppler) and five malignant features (irregular solid tumor, irregular multilocular solid tumor with largest diameter ≥100 mm, presence of ascites, ≥4 papillary projections, very strong vascularization on color Doppler); (2) Simple rules risk: risk model based on logistic regression using the 10 binary features used in the simple rules and type of center (oncology center vs. other); (3) LR2: risk model based on logistic regression based on age (years), presence of acoustic shadows, presence of ascites, presence of papillary projections with blood flow, maximum diameter of largest solid component, irregular internal cyst walls, and (4) The ADNEX model uses nine predictors, including three clinical variables (age, serum CA-125 level, and type of center: oncology referral center vs. other), and six ultrasound features, maximal diameter, proportion of solid tissue, more than 10 cyst locules, number of papillary projections, acoustic shadows, and ascites. All are highly effective but have their own advantages and disadvantages. Though we did not directly compare the performance of modified RMI to that of the other models mentioned above, it might be assumed that the performances in discriminating between benign and malignant are comparable. For example, the AUC of the ADNEX model for the classic discrimination between benign and malignant tumors was 0.94 (0.93 to 0.95) [17], whereas the modified RMI in this study gave an AUC of 0.931. RMI and IOTA simple rule may be suitable for clinical use in primary health care centers by general practitioners [8,11,12]. The IOTA simple rule seems to be more complicated in practice, and approximately 20% of examinations give inconclusive results [8,11,19]. The main disadvantage of RMI is the requirement for the measurement of CA-125 levels. From our point of view, in developing countries with limited resources, IOTA simple rule or modified RMI by general gynecologists or non-expert examiners can be a method of choice in the evaluation of adnexal masses. Both have different individual advantages and disadvantages, which must be taken into account for clinical use. RMI is simpler in practice but needs the measurement of serum CA-125 levels, whereas the IOTA simple rule solely depends on ultrasound features and is relatively more complicated and is associated with inconclusive results, which necessitates expert sonographer or oncologist consultation or referral to a tertiary care center.

The strengths of our study include: (1) Each comparison of the effectiveness of the three methods was performed on the same patient by the same examiner and in the same ultrasound settings, leading to a perfect comparison. (2) The study had an adequate sample size to gain the power to express the significance of a small difference, if it existed, in comparison. (3) The ultrasound examiners were general gynecologists, not experts on gynecologic ultrasound. Accordingly, the results of this study are likely reproduced in actual practice by general practitioners.

The limitations of this study include: (1) Intra- and inter-observer variations in interpretation of sonographic features were not evaluated. (2) Selection bias might have existed because only the cases scheduled for surgery were included, whereas the cases treated expectantly were not enrolled. The results of this study represent only this group of cases, in which the prevalence of malignant tumors was higher than that in the general population. Accordingly, the positive and negative predictive value should be interpreted with caution, because the predictive value depends on the prevalence, which is higher in the selected group than that in the general population. Nevertheless, the sensitivity and specificity, as well as ROC curve, which are used to evaluate the validity of the test, are independent from the prevalence and still reliably represent the performance of test. Additionally, the cases not scheduled for surgery are usually benign or functional and need only follow-up. In actual practice, the need for differentiation is usually confined to the group with scheduled surgery.

RMI is commonly used by many general gynecologists because of its simplicity, use of parameters which are assessed in daily practice and the lack of a need for any electronic application for risk assessment other than simple manual calculation. Since modified RMI has better performance, it should be considered to replace the conventional RMI. Nevertheless, further studies by different groups of researchers are needed to confirm the reproducibility as external validation. On ultrasound examination to assess adnexal mass, we encourage general gynecologists or practitioners, who are already familiar to conventional RMI, to pay more attention to the following features: (1) moderate to strong color flow (color score 3–4), (2) presence of papillary projections or nodules arising from the cyst wall or the septum, (3) solid area, (4) ascites and (5) intra-abdominal metastases.

## 5. Conclusions

This study demonstrated that the modified RMI had better diagnostic performance than conventional RMI in distinguishing malignant from benign adnexal masses. Because of its higher effectiveness, simplicity and lack of a need for high expertise, the modified RMI may be applied in daily practice by general gynecologists or practitioners to triage patients with adnexal masses, especially in primary health care centers.

## Figures and Tables

**Figure 1 ijerph-20-00888-f001:**
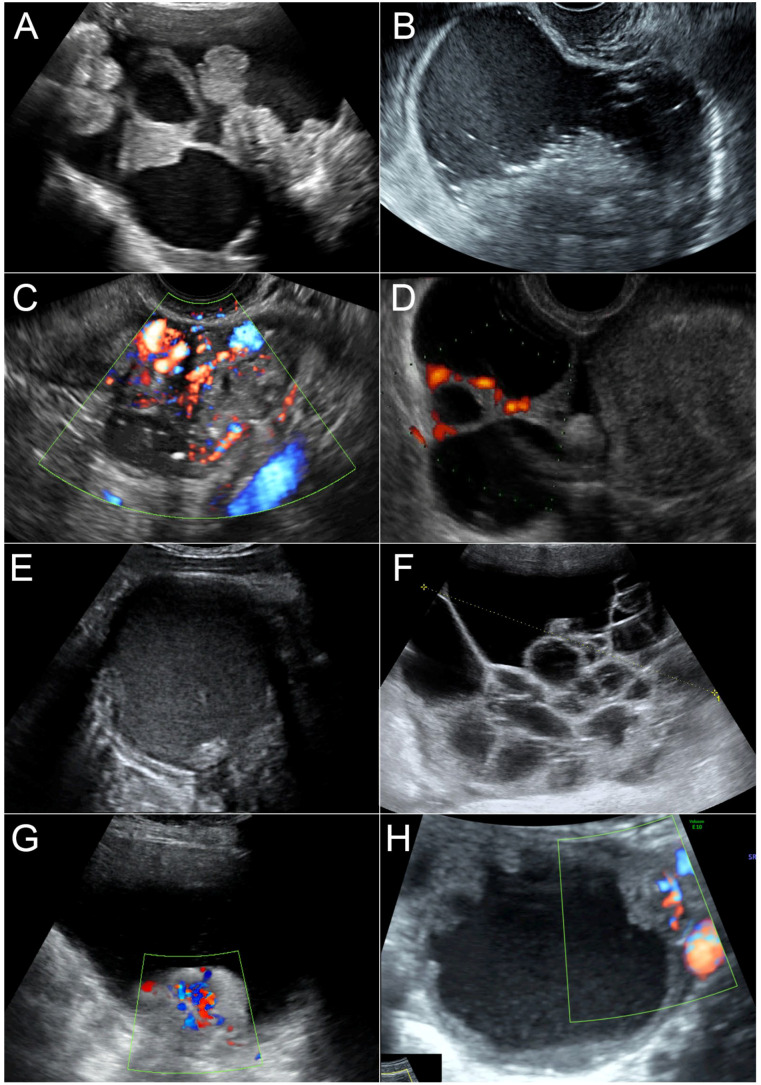
Examples of ultrasound features of adnexal masses; (**A**) serous cystadenocarcinoma: multilocular cystic mass with solid part arising from the septae; (**B**) cystic mature teratoma: cystic mass with solid heterogeneous components; (**C**) metastatic carcinoma: solid mass with high vascularity; (**D**) endometrioid carcinoma: solid-cystic mass with vascularization in the septae; (E) endometrioma: unilocular cyst with ground-glass appearance and a single small nodule; (**F**) mucinous cystadenoma: multiple daughter cysts; (**G**) clear cell carcinoma: cystic mass with solid part arising from the wall with high vascularity; (**H**) serous low-malignant-potential tumor: unilocular cyst with multiple papillary projections arising from the wall.

**Figure 2 ijerph-20-00888-f002:**
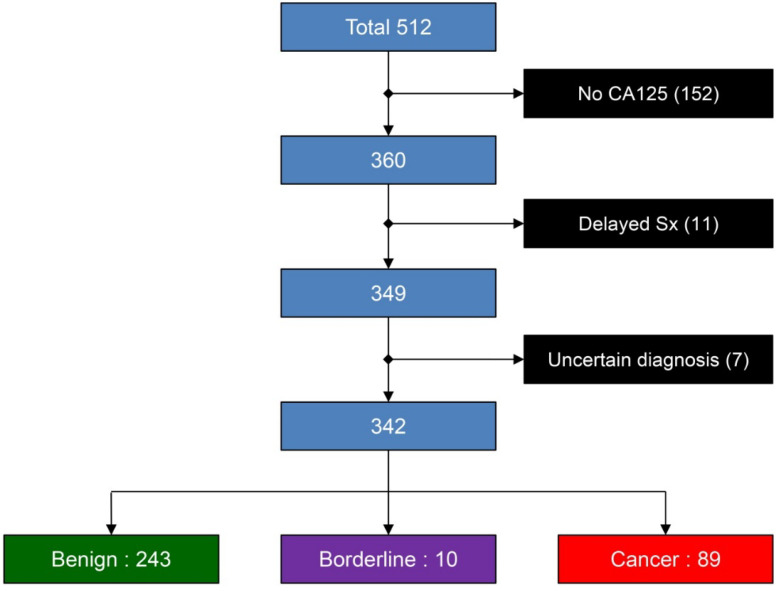
Flowchart of the cases scheduled for surgery because of adnexal masses.

**Figure 3 ijerph-20-00888-f003:**
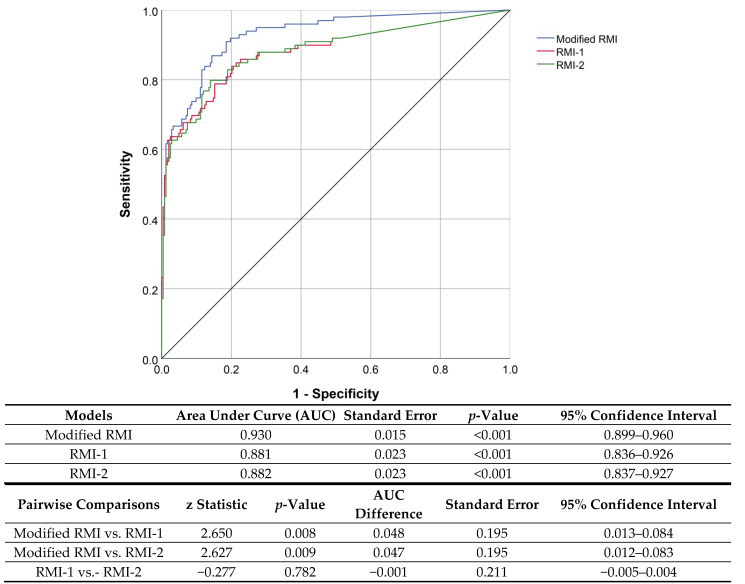
Receiver operating characteristic (ROC) curves present diagnostic performance of the three systems of risk of malignancy index (RMI) in predicting ovarian malignancy, together with the tables of AUC comparisons among the three systems (area under the curve; AUC: 0.930, 0.881 and 0.882 for modified RMI, RMI-1 and RMI2, respectively).

**Table 1 ijerph-20-00888-t001:** Frequencies of the adnexal masses according to final diagnosis.

Tumor Groups	Final Diagnoses	Number	Percent
Benign tumors		**243**	**71.1**
	Endometrioma	80	23.4
	Mature teratoma	47	13.7
	Serous cyst	26	7.6
	Mucinous cyst	21	6.1
	Pedunculated leiomyoma	20	5.8
	Pseudocyst	20	5.8
	Simple epithelial cyst	9	2.6
	Fibroma	8	2.3
	Tubo-ovarian abscess	6	1.8
	Hemorrhagic mass	3	0.9
	Brenner tumor	3	0.9
Low malignant potential (LMP)		**10**	**2.9**
	Mucinous tumor of LMP	6	1.8
	Serous tumor of LMP	4	1.2
Malignant tumors		**89**	**26.0**
	Endometrioid adenocarcinoma	25	7.3
	Serous adenocarcinoma	22	6.4
	Mucinous adenocarcinoma	20	5.8
	Clear cell carcinoma	7	2.0
	Immature teratoma	4	1.2
	Dysgerminoma	4	1.2
	Sex-cord stromal tumor	3	0.9
	Yolk sac tumor	2	0.6
	Metastatic cancer	2	0.6
**Total**		**342**	**100.0**

**Table 2 ijerph-20-00888-t002:** Diagnostic performance of risk of malignancy index (RMI) in predicting malignant ovarian masses.

	Malignant (99)	Benign (243)	Sens. % (95% CI)	Spec. % (95% CI)	PPV % (95% CI)	NPV % (95% CI)
Modified RMI						
High scores	87	43	87.9 (81.4–94.3)	82.3 (77.5–87.1)	66.9 (58.8–75.0)	94.3 (90.4–98.3)
Low scores	12	200				
RMI-1						
High scores	74	38	74.7 (66.2–83.3)	84.4 (79.8–88.9)	66.1 (57.3–74.8)	89.1 (83.4–94.9)
Low scores	25	205				
RMI-2						
High scores	79	46	79.8 (71.9–87.7)	81.1 (76.1–86.0)	63.2 (54.7–71.7)	90.8 (85.7–95.9)
Low scores	20	197				

Sens.: sensitivity; Spec.: specificity; PPV: positive predictive value; NPV: negative predictive value; CI: confident interval.

## Data Availability

The datasets analyzed during the current study are available from the corresponding author upon reasonable request.

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
