# Peer review of "Comparisons of Effectiveness in Differentiating Benign from Malignant Ovarian Masses between Conventional and Modified Risk of Malignancy Index (RMI)"

_ijerph, 2023, doi:10.3390/ijerph20010888_

Round 1
Reviewer 1 Report (Previous Reviewer 1)
Dear Authors,
my comments:
1. ALL abbrevations in text and tables should be described.
2. All tables and figures needs legends if you use abbreviatons needs legends.
3. Comparison of IOTA and RMI is not fully described. IOTA is not just "simple rules", but also ADNEX model, where Ca125 and several other parameters are used to asses the risk of malignancy.
Author Response
Reviewer 1
Comments and Suggestions for Authors
Dear Authors,
my comments:
- ALL abbrevations in text and tables should be described.
Response: In revised MS, all full-text terms of abbreviations are provided.
- All tables and figures needs legends if you use abbreviatons needs legends.
Response: In revised MS, all table and figures have the legends. (Note that Figure 3 includes addendum tables of AUC comparisons to illustrate the above ROC curves. The tables are part of Figure 3)
- Comparison of IOTA and RMI is not fully described. IOTA is not just "simple rules", but also ADNEX model, where Ca125 and several other parameters are used to asses the risk of malignancy.
Response: In revised MS, we describe more about other models (IOTA, LR1, LR2, ADNEX model and comparison with our findings, as highlighted in “Discussion” page 8-9.

Reviewer 2 Report (Previous Reviewer 3)
Thank you for requesting to provide a review of this revised article, which has a subject of high interest
The main aim of the study was to compare the predictive performance in differentiating benign from malignant ovarian masses between modified risk malignant index (RMI) and conventional RMI (RMI-1 and RMI-2). The study was conducted as a secondary analysis and the popuation included in the analysis were patients admitted for elective pelvic operation because of adnexal mass, for a period of time between April 2016 and March 2022, which is sufficient for such a study.
Regarding the structure and accuracy of the phrases, the manuscript has well structured information, and the phrases are well understandable.
The manuscript is original and well defined.
The results provide an advance in current knowledge.
The results are being interpreted appropriately and are significant. The conclusions are consistent with the evidence and the arguments presented, and they adress properly to the main question which conducted the analysis.
The references are appropriate and well suited for this kind of study.
The data are robust enough to draw the conclusion.
Surely the paper will attract a wide readership.
The English language is appropriate and has only a few writting mistakes, which need to be corrected so that the article can be of highest quality.
It is clear that the article should be published.
Author Response
Reviewer 2
Comments and Suggestions for Authors
Thank you for requesting to provide a review of this revised article, which has a subject of high interest
The main aim of the study was to compare the predictive performance in differentiating benign from malignant ovarian masses between modified risk malignant index (RMI) and conventional RMI (RMI-1 and RMI-2). The study was conducted as a secondary analysis and the popuation included in the analysis were patients admitted for elective pelvic operation because of adnexal mass, for a period of time between April 2016 and March 2022, which is sufficient for such a study.
Regarding the structure and accuracy of the phrases, the manuscript has well structured information, and the phrases are well understandable.
The manuscript is original and well defined.
The results provide an advance in current knowledge.
The results are being interpreted appropriately and are significant. The conclusions are consistent with the evidence and the arguments presented, and they adress properly to the main question which conducted the analysis.
The references are appropriate and well suited for this kind of study.
The data are robust enough to draw the conclusion.
Surely the paper will attract a wide readership.
The English language is appropriate and has only a few writting mistakes, which need to be corrected so that the article can be of highest quality.
Response: In revised MS, English has been reviewed and corrected.
It is clear that the article should be published.
Response: Thank you very much.

Round 2
Reviewer 1 Report (Previous Reviewer 1)
Dear Authors,
I accept your reply.
This manuscript is a resubmission of an earlier submission. The following is a list of the peer review reports and author responses from that submission.
Round 1
Reviewer 1 Report
Dear Authors,
These results could be interesting.
I hope you will perform further studies about this topic
One question:
1. What is AROMA? line 197
Reviewer 2 Report
Dear author's
I was pleased to review your article and I have the following comments:
1. Please explain the novelty of your study.
2. Introduction should introduce the reader with the general concept of the research.
3. Please specify the US protocol, considering that in the study the the US was performed by multiple gynaecologists.
4 The section Discussion in the part of the article that compare your results with the existing literature. Please add references in the section Discussion.
5. R 201-204 I suggest you to read/cite the article https://link.springer.com/article/10.1007/s12262-020-02455-w
Reviewer 3 Report
Thank you for requesting to provide a review of this article, which has a subject of high interest, as differentiating benign from malignant ovarian masses is one of the most important aspects for the surgical treatment, due to the fact that surgery cand be either radical, or can preserve the ovaries, especially if the patient is young.
The main aim of the study was to compare the predictive performance in differentiating benign from malignant ovarian masses between modified risk malignant index (RMI) and conventional RMI (RMI-1 and RMI-2). The study was conducted as a secondary analysis and the popuation included in the analysis were patients admitted for elective pelvic operation because of adnexal mass, for a period of time between April 2016 and March 2022, which is sufficient for such a study.
Regarding the structure and accuracy of the phrases, the manuscript has well structured information, and the phrases are well understandable.
The manuscript is original and well defined.
The results provide an advance in current knowledge.
The results are being interpreted appropriately and are significant.
The data are robust enough to draw the conclusion.
Surely the paper will attract a wide readership.
The English language is appropriate and has only a few writting mistakes, which need to be corrected so that the article can be of highest quality.
There are only a few things to be added in the lines below, but it is clear that the article should be published:
Line 31: „,” after „counseling”
Line 34: safely selected for laparoscopic, not „safely selected for by laparoscopic”
Line 41: had a sensitivity of 85%, not „had a sensitivity was 85%”
Line 54: „,” after „prediction”
Line 74: to participate in the project, not „to participate with the project”
Line 198: we included, not „we include”
Line 205: with the intention, not „with intention”